# Wasserstein Distances for Stereo Disparity Estimation

**Divyansh Garg**[1]   **Yan Wang**[1]   **Bharath Hariharan**[1]
**Mark Campbell**[1]   **Kilian Q. Weinberger**[1]   **Wei-Lun Chao**[2]
[1]Cornell University, Ithaca, NY    [2]The Ohio State University, Columbus, OH
{dg595, yw763, bh497, mc288, kqw4}@cornell.edu    chao.209@osu.edu

## Abstract

Existing approaches to depth or disparity estimation output a distribution over a set of pre-defined discrete values. This leads to inaccurate results when the true depth or disparity does not match any of these values. The fact that this distribution is usually learned indirectly through a regression loss causes further problems in ambiguous regions around object boundaries. We address these issues using a new neural network architecture that is capable of outputting arbitrary depth values, and a new loss function that is derived from the Wasserstein distance between the true and the predicted distributions. We validate our approach on a variety of tasks, including stereo disparity and depth estimation, and the downstream 3D object detection. Our approach drastically reduces the error in ambiguous regions, especially around object boundaries that greatly affect the localization of objects in 3D, achieving the state-of-the-art in 3D object detection for autonomous driving. Our code will be available at https://github.com/Div99/W-Stereo-Disp.

## 1   Introduction

Depth estimation from stereo images is a long-standing task in computer vision [28, 34]. It is a key component of many downstream problems, ranging from 3D object detection in autonomous vehicles [8, 19, 31, 40, 51] to graphics applications such as novel view generation [21, 52]. The importance of this task in practical applications has led to a flurry of recent research. Convolutional networks have now superseded more classical techniques and led to significant improvements in accuracy [4, 25, 41, 54].

These techniques estimate depth by finding accurate pixel correspondences and estimating the *disparity* between their $X$-coordinates, which is inversely proportional to depth. Because pixels have integral coordinates, so does the estimated disparity — causing even the resulting depth estimates to be discrete. This introduces inaccuracy, as the ground truth disparity and depth are

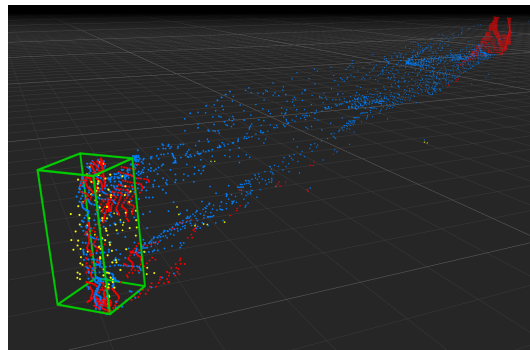

Figure 1: **The effect of our continuous disparity network (CDN).** We show a person (green box) in front of a wall. The blue 3D points are obtained using PSMNet [4]. The red points from our **CDN** model are much better aligned with the shape of the objects: they do not suffer the streaking artifacts near edges. Yellow points are from the ground truth LiDAR. (One floor square is 1m×1m.)

naturally real-valued. This discrepancy is typically addressed by predicting a *categorical distribution* over a fixed set of discrete values, and then computing the *expected* depth from this distribution, which can in theory be any arbitrary real value (within the range of the set) [4, 12, 41, 51, 54].

In this paper, we argue that such a design choice may lead to inaccurate depth estimates, especially around object boundaries. For example, in Figure 1 we show the pixels (back-projected into 3D

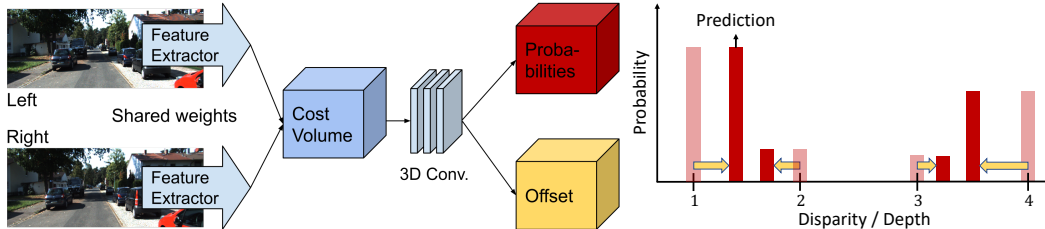

Figure 2: **Continuous disparity network (CDN).** We propose to predict a *real-value* offset (yellow arrows) for each pre-defined discrete disparity value (e.g., $\{1, 2, 3, 4\}$), turning a categorical distribution (magenta bars) to a continuous distribution (red bars), from which we can output the mode disparity for accurate estimation.

using the depth estimates) along the boundary between a person in the foreground at 30m depth and a wall in the background at 70m depth. The predicted depth distribution of these border pixels is likely to be multi-modal, having two peaks around 30 and 70 meters. Simply taking the mean outputs a low probability value in between the two modes (e.g., 50m). Such "smoothed" depth estimates can have a strong negative impact on subsequent 3D object detection, as they "smear" the pedestrian around the edges towards the background (note the many blue points between the wall and the pedestrian). A bounding box including all these trailing points, far from the actual person, would strongly misrepresent the scene's geometry. What may further aggravate the problem is how the distribution is usually learned. Existing approaches mostly learn the distribution via a regression loss: minimizing the distance between the mean value and the ground truth [12, 51]. In other words, there is no direct supervision to teach the model to assign higher probabilities around the truth depth.

To address these issues, we propose a novel neural network architecture for stereo disparity estimation that is capable of outputting a distribution over *arbitrary* disparity values, from which we can directly take the mode and bypass the mean. As with existing work, our model predicts a probability for each disparity value in a pre-defined, discrete set. Additionally, it predicts a real-valued *offset* for each discrete value. This is a simple architectural modification, but it has a profound impact. With these offsets, the output is converted from a discrete categorical distribution to a *continuous* distribution over disparity values: a mixture of Dirac delta functions, centered at the pre-defined discrete values shifted by predicted offsets[1]. This simple addition of predicted offsets allows us to use the mode as the prediction during inference, instead of the mean, guaranteeing that the predicted depth has a high estimated probability. Figure 2 illustrates our model, **continuous disparity network (CDN)**.

Next, we propose a novel loss function that provides a more informative objective during training. Concretely, we allow uni- or multi-modal ground truth depth distributions (obtained from nearby pixels) and represent them as (mixtures of) Dirac delta functions. The learning objective is then to minimize the divergence between the predicted and the ground truth distributions. Noting that the two distributions might not have a common support, we apply the Wasserstein distance [39] to measure the divergence. While computing the exact Wasserstein distance of arbitrary distributions can be time-consuming, computing it for one-dimensional distributions (e.g., distributions of one-dimensional disparity) enjoys efficient solutions, creating negligible training overhead.

Our proposed approach is both mathematically well-founded and practically extremely simple. It is compatible with most existing stereo depth or disparity estimation approaches — we only need to add an additional offset branch and replace the commonly used regression loss by the Wasserstein distance. We validate our approach using multiple existing stereo networks [4, 51, 54] on three tasks: stereo disparity estimation [25], stereo depth estimation [9], and 3D object detection [9]. The last is a downstream task using stereo depth as the input to detect objects in 3D. We conduct comprehensive experiments and show that our algorithm leads to significant improvement in all three tasks.

## 2 Background

Stereo techniques rely on two cameras oriented parallel and translated horizontally relative to each other [46, 53]. In this setting, for a pixel $(u, v)$ in one image, the corresponding pixel in the second

image is constrained to be at $(u + D(u, v), v)$, where $D(u, v)$ is called the *disparity* of the pixel. The disparity is inversely proportional to the *depth* $Z(u, v) : D(u, v) = \frac{f \times b}{Z(u,v)}$, where $b$ is the translation between the cameras (called the *baseline*) and $f$ is the focal length of the cameras. Stereo depth estimation techniques typically first estimate disparity in units of pixels and then exploit the reciprocal relationship to approximate depth. The basic approach is to compare pixels $(u, v)$ in the left image $I_l$ with pixels $(u, v + d)$ in the right image $I_r$ for different values of $d$, and find the best match. Since pixel coordinates are constrained to be integers, $d$ is constrained to be an integer as well. The estimated disparity is thus an integer, forcing the estimated depth to be one of a few discrete values.

Instead of producing a single integer-valued disparity value, modern pipelines produce a *distribution* over these possible disparities [4, 12]. They do this by constructing a 4D *disparity feature volume*, $C_{\text{disp}}$, in which $C_{\text{disp}}(u, v, d, :)$ is a feature vector that captures the difference in appearance between $I_l(u, v)$ and $I_r(u, v + d)$. This feature vector can be, for instance, the concatenation of the feature vectors of the two pixels, in turn obtained by running a convolutional network on each image. The disparity feature volume is then passed through a series of 3D convolutional layers, culminating in a cost for each disparity value $d$ for each pixel, $S_{\text{disp}}(u, v, d)$ [4]. By taking $\text{softmax}$ along the disparity dimension, one can turn $S_{\text{disp}}(u, v, d)$ into a probability distribution [24]. Because we only consider integral disparity values, this distribution is a categorical distribution over the possible disparity values (e.g., $d \in \{0, \cdots, 191\}$). One can then obtain the disparity $D(u, v)$, for example, by $\text{argmax}_d \, \text{softmax}(-S_{\text{disp}}(u, v, d))$. However, in order to obtain continuous disparity estimates beyond integer-valued disparities, [4, 12, 41, 54] apply the following weighted combination (i.e., mean),

$$D(u, v) = \sum_d \text{softmax}(-S_{\text{disp}}(u, v, d)) \times d. \tag{1}$$

The whole neural network can be learned end-to-end, including the image feature extractor and 3D convolution kernels, to minimize the disparity error (on one image)

$$\sum_{(u,v) \in \mathcal{A}} \ell(D(u, v) - D^\star(u, v)), \tag{2}$$

where $\ell$ is the smooth L1 loss, $D^\star$ is the ground truth map, and $\mathcal{A}$ contains pixels with ground truths.

Recently, [51] argue that learning with Equation 2 may over-emphasize nearby depths, and accordingly propose to learn the network directly to minimize the depth loss. Specifically, they constructed depth cost volume $S_{\text{depth}}(u, v, z)$, rather than $S_{\text{disp}}(u, v, d)$, and predicted the continuous depth by

$$Z(u, v) = \sum_z \text{softmax}(-S_{\text{depth}}(u, v, z)) \times z. \tag{3}$$

The entire network is learned to minimize the distance to the ground truth depth map $Z$

$$\sum_{(u,v) \in \mathcal{A}} \ell(Z(u, v) - Z^\star(u, v)). \tag{4}$$

In this paper, we argue that the design choices to output continuous values (Equation 1 and Equation 3) can be harmful to pixels in ambiguous regions, and the objective functions for learning the networks (Equation 2 and Equation 4) do not directly match the predicted distribution to the true one. The most similar work to ours is [55], which learns the network with a distribution matching loss on $\text{softmax}(-S_{\text{disp}}(u, v, d))$; however, they still need to apply Equation 1 to obtain continuous estimates. Luo et al. [24] also learned the network by distribution matching, but applied post-processing (e.g., semi global block matching) to obtain continuous estimates.

**Stereo-based 3D object detection.** 3D object detection has attracted significant attention recently, especially for the application of self-driving cars [3, 5, 9, 10, 13, 37]. While many algorithms rely on the expensive LiDAR sensor as input [15, 16, 30, 35, 47], several recent papers have shown promising accuracy using the much cheaper stereo images [8, 14, 17, 19, 29, 43, 45]. One particular framework is Pseudo-LiDAR [31, 40, 51], which converts stereo depth estimates into a 3D point cloud that can be inputted to any existing LiDAR-based detector, achieving the state-of-the-art results.

## 3 Disparity Estimation

*For brevity, in the following we mainly discuss disparity estimation. The same technique can easily be applied to depth estimation, which is usually adapted from their disparity estimation counterparts.*

As reviewed in section 2, many existing stereo networks output a distribution of disparities at each pixel. This distribution is a categorical distribution over discrete disparity values: discrete because they are estimated as the difference in $X$-coordinates of corresponding pixels, and as such are integers. Stereo techniques then compute the mean of the distribution to obtain a continuous estimate that is not limited to integral values.

We point out two disadvantages of taking the mean estimate. First, the mean value can deviate from the mode and may wrongly predict values of low probability when the predicted distribution is multi-modal (see Figure 3). Such multi-modal distributions appear frequently at pixels around the object bound-

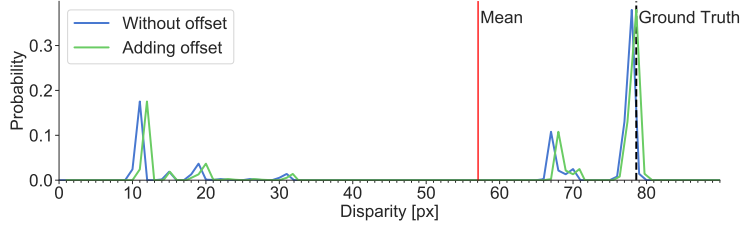

Figure 3: **The predicted disparity posterior for a pixel on object boundaries.** The uni-modal assumption can break down, leading to a *mean* estimate that is in a low probability region. Learning offsets allow us to predict the continuous *mode*. (Offsets are in $[0, 1]$ here.)

aries. While they collectively occupy only a tiny portion of image pixels, recent studies have shown their particular importance in the downstream tasks like 3D object detection [18, 19, 31]. For instance, let us consider a street scene where a car 30m away (a disparity of, say, 10 pixels) is driving on the road towards the camera, with the sky as the background. The pixels on the car boundary can either take a disparity of around 10 pixels (for the car) or a disparity of 0 pixels (for the sky). Simply taking the mean likely produces arbitrary disparity estimates between these values, producing depth estimates that are neither on the car nor on the background. The downstream 3D object detector can, therefore, wrongly predict the car orientation and size, potentially leading to accidents. Second, the physical meaning of the mean value is by no means aligned with the true disparity: uncertainty in correspondence might yield a $40\%$ chance of a disparity of 10 pixels and a $60\%$ chance for a disparity of 20 pixels, but this does not mean that the disparity should be 16 pixels.

Instead, a more straightforward way to simultaneously model the uncertainty and output continuous disparity estimates is to extend the support of the output distribution beyond integers.

### 3.1 Continuous disparity network (CDN)

To this end, we propose a new neural network architecture and output representation for disparity estimation. The output of our network will still be a set of discrete values with corresponding probabilities, but the discrete values *will not be restricted to integers*. The key idea is to start with integral disparity values, and predict *offsets* in addition to probabilities.

Denote by $\mathcal{D}$ the set of integral disparity values. As above, disparity estimation techniques produce a cost $S_{\text{disp}}(u, v, d)$ for every $d \in \mathcal{D}$. A softmax converts this cost into a probability distribution:

$$p(d|u, v) = \begin{cases} \text{softmax}(-S_{\text{disp}}(u, v, d)) & \text{if } d \in \mathcal{D}, \\ 0 & \text{otherwise.} \end{cases} \quad (5)$$

We propose to add a sub-network $b(u, v, d)$ that predicts an offset disparity value for each integral disparity value $d \in \mathcal{D}$ at each pixel $(u, v)$. We use this to displace the probability mass at $d \in \mathcal{D}$ to $d' = d + b(u, v, d)$. This results in the following probability distribution:

$$\tilde{p}(d'|u, v) = \sum_{d \in \mathcal{D}} p(d|u, v)\delta(d' - (d + b(u, v, d))), \quad (6)$$

which is a mixture of Dirac delta functions over arbitrary disparity values $d'$. In other words, $\tilde{p}$ has $|\mathcal{D}|$ supports, each located at $d + b(u, v, d)$ with a weight $p(d|u, v)$. The resulting continuous disparity estimate $D(u, v)$ at $(u, v)$ is the **mode** of $\tilde{p}(d'|u, v)$.

Our network design with a sub-network for offset prediction is reminiscent of G-RMI pose estimator [28] and one-stage 2D object detectors [20, 22, 33]. The former predicts the heatmaps (at fixed locations) and offsets for each keypoint; the latter parameterizes the predicted bounding box coordinates by the anchor box location plus the predicted offset. One may also interpret our approach as a coarse-to-fine depth prediction, first picking the bin centered around $\text{argmax}_{d \in \mathcal{D}} \, p(d|u, v)$ and then locally adjusting it by an offset.

In our implementation, the sub-network $b(u, v, d)$ shares its feature and computation with $S_{\text{disp}}(u, v, d)$ except for the last block of fully-connected or convolutional layers.

## 3.2 Learning with Wasserstein distances

We propose to train our disparity network such that the mixture of Dirac delta functions (Equation 6) is directly learned to match the ground truth distribution. Concretely, we represent the distribution of ground truth disparity at a pixel $(u, v)$, $p^\star(d'|u, v)$, as a Dirac delta function centered at the ground truth disparity $d^\star = D^\star(u, v)$: $p^\star(d'|u, v) = \delta(d' - d^\star)$. We then employ a learning objective to minimize the divergence (distance) between $\tilde{p}(d'|u, v)$ and $p^\star(d'|u, v)$. There are many popular divergence measures between distributions, such as Kullback-Leibler divergence, Jensen-Shannon divergence, total Variation, the Wasserstein distance, etc. In this paper, we choose the Wasserstein distance for one particular reason: $\tilde{p}(d'|u, v)$ and $p^\star(d'|u, v)$ may not have any common supports.

The Wasserstein-$p$ distance between two distributions $\mu, \nu$ over a metric space $(X, d)$ is defined as

$$W_p(\mu, \nu) = \left( \inf_{\gamma \in \Gamma(\mu, \nu)} \mathbb{E}_\gamma \, d(x, y)^p \right)^{1/p}, \tag{7}$$

where $\Gamma(\mu, \nu)$ denotes the set of all the joint distributions $\gamma(x, y)$ whose marginal distributions $\gamma(x)$ and $\gamma(y)$ are exactly $\mu$ and $\nu$, respectively. Intuitively, $\gamma(x, y)$ indicates how much "mass" to be transported from $x$ to $y$ in order to transform the distribution $\mu$ to $\nu$.

Estimating the Wasserstein distance is usually non-trivial and requires solving a linear programming problem. One particular exception is when $\mu$ and $\nu$ are both distributions of one-dimensional variables, which is the case for our distribution over disparity values[2]. Specifically, when $\nu$ is a Dirac delta function whose support is located at $y^\star$, the Wasserstein-$p$ distance can be simplified as

$$W_p(\mu, \nu) = (\mathbb{E}_\mu \, \mathbb{E}_\nu \, \|x - y\|^p)^{1/p} = (\mathbb{E}_\mu \, \|x - y^\star\|^p)^{1/p}. \tag{8}$$

By plugging $\tilde{p}(d'|u, v)$ and $p^\star(d'|u, v)$ into $\mu$ and $\nu$ respectively, we obtain

$$W_p(\tilde{p}, p^\star) = (\mathbb{E}_{\tilde{p}} \, \|d' - d^\star\|^p)^{1/p} = \left( \sum_{d \in \mathcal{D}} p(d|u, v) \, \|d + b(u, v, d) - d^\star\|^p \right)^{1/p} \tag{9}$$

$$= \left( \sum_{d \in \mathcal{D}} \text{softmax}(-S_{\text{disp}}(u, v, d)) \, \|d + b(u, v, d) - d^\star\|^p \right)^{1/p},$$

based on which we can learn the conventional disparity network (red) and the additional offset sub-network (blue) jointly (i.e., by minimizing Equation 9). We focus on $W_1$ and $W_2^2$ distances.

## 3.3 Extension: learning with multi-modal ground truths

One particular advantage of learning to match the distributions is the capability of allowing multiple ground truth values (i.e., a multi-modal ground truth distribution) at a single pixel location. Denote $\mathcal{D}^\star$ as the set of ground truth disparity values at a pixel $(u, v)$, the ground truth distribution becomes

$$p^\star(d'|u, v) = \sum_{d^\star \in \mathcal{D}^\star} \frac{1}{|\mathcal{D}^\star|} \delta(d' - d^\star). \tag{10}$$

Since $p^\star(d'|u, v)$ is not a Dirac delta function, we can no longer apply Equation 8 but the following equation for comparing two one-dimensional distributions [27, 32, 42]

$$W_p(\tilde{p}, p^\star) = \left( \int_0^1 \left| \tilde{P}^{-1}(x) - P^{\star-1}(x) \right|^p \mathrm{d}x \right)^{1/p}, \tag{11}$$

where $\tilde{P}$ and $P^\star$ are the cumulative distribution functions (CDFs) of $\tilde{p}$ and $p^\star$, respectively. For the case $p = 1$, we can rewrite Equation 11 as [38]

$$W_1(\tilde{p}, p^\star) = \int_{\mathbb{R}} \left| \tilde{P}(d') - P^\star(d') \right| \mathrm{d}d'. \tag{12}$$

We note that, both Equation 11 and Equation 12 can be computed efficiently.

While existing datasets do not provide multi-modal ground truths directly, we investigate the following procedure to construct them. For each pixel, we consider a $k \times k$ neighborhood and create a multi-modal distribution by setting the center-pixel disparity with a weight $\alpha$ and the remaining ones each with $\frac{1-\alpha}{k \times k - 1}$. We set $k = 3$ and $\alpha = 0.8$ in the experiment. Our empirical study shows that using a multi-modal ground truth leads to a much faster model convergence.

### 3.4 Comparisons to related work

Kendall et al. [12] discussed the use of means or modes. They employed pre-scaling to sharpen the predicted probability, which might resolve the multi-modal issue but makes the prediction concentrate on discrete disparity values. In contrast, we do not prevent predicting a multi-modal distribution, especially for pixels whose disparities are inherently multi-modal. We output the mode (after an offset), which is what Kendall et al. [12] hoped to achieve. We note that 3D convolutions can smooth the estimation but cannot guarantee uni-modal distributions.

Compared to G-RMI pose estimator and one-stage 2D object detectors mentioned in subsection 3.1, our work learns the two (sub-)networks jointly using a single objective function rather than a combination of two separated ones. See the supplementary material for more comparisons. Liu et al. [23] propose to use the Wasserstein loss for pose estimation to characterize the inter-class correlations; however, they do not predict offsets for pre-defined discrete pose labels. Our work is also related to [2], in which the authors propose to learn the value distribution, instead of the expected value, using the Wasserstein loss for reinforcement learning.

## 4 Experiments

### 4.1 Datasets and metrics

**Datasets.** We evaluate our method on two challenging stereo benchmark datasets, i.e., Scene Flow [25] and KITTI 2015 [26], and on a 3D object detection benchmark KITTI 3D [9, 10].

**1) Scene Flow [25].** Scene Flow is a large synthetic dataset containing 35,454 training image pairs and 4,370 testing image pairs, where the ground truth disparity maps are densely provided, which is large enough for directly training deep neural networks.

**2) KITTI 2015 [26].** KITTI 2015 is a real-world dataset with street scenes captured from a driving car. KITTI 2015 contains 200 training stereo image pairs with sparse ground truth disparities obtained using LiDAR, and 200 testing image pairs with ground truth disparities held by evaluation server for submission evaluation only. Its small size makes it a challenging dataset.

**3) KITTI 3D [9, 10].** KITTI 3D contains 7,481 (pairs of) images for training and 7,518 (pairs of) images for testing. We follow the same training and validation splits as suggested by Chen et al. [6], containing 3,712 and 3,769 images, respectively. For each image, KITTI provides the corresponding Velodyne LiDAR point cloud (for sparse depth ground truths), camera calibration matrices, and 3D bounding box annotations. We evaluate our approach by plugging it into existing stereo-based 3D object detectors [8, 40, 51], which all require stereo depth estimation as a key component.

**Metrics.** We evaluate our methods on three tasks: stereo disparity estimation, stereo depth estimation, and 3D object detection. We apply the corresponding standard metrics listed as follows.

**1) stereo disparity**, we use two standard metrics: End-Point-Error (EPE), i.e., the average difference of the predicted disparities and their true ones, and $k$-Pixel Threshold Error (PE), i.e., the percentage of pixels for which the predicted disparity is off the ground truth by more than $k$ pixels. We use the 1-pixel and 3-pixel threshold errors, denoted as 1PE and 3PE. PE is robust to outliers with large disparity errors, while EPE measures errors to sub-pixel level.

**2) stereo depth.** We use the Root Mean Square Error (RMSE) $\sqrt{\frac{1}{|\mathcal{A}|} \Sigma_{(u,v) \in \mathcal{A}} |z(u,v) - z^\star(u,v)|^2}$ and Absolute Relative Error (ABSR) $\frac{1}{|\mathcal{A}|} \Sigma_{(u,v) \in \mathcal{A}} \frac{|z(u,v) - z^\star(u,v)|}{z^\star(u,v)}$, where $\mathcal{A}$ denotes all the pixels having ground truths, and $z$ and $z^\star$ are estimated depth and ground truth depth respectively.

**3) 3D object detection.** We focus on 3D and bird's-eye-view (BEV) localization and report the results on the official leader board and the validation set. Specifically, we focus on the "car" category,

Table 1: **Disparity results**. We report results on Scene Flow and KITTI 2015. For Scene Flow, end point errors (EPE) and the 1-pixel and 3-pixel threshold error rates (1PE, 3PE) are reported. For KITTI 2015 we report the standard metrics (using 3PE) for both Non-occluded and All pixels regions. Methods based on **CDN** are highlighted in blue. Lower is better. The best result per column in in bold. *Since the baselines are mostly trained with uni-modal ground truths, we only show* **CDN** *with the same ground truths here for a fair comparison.*

| Method | Scene Flow | | | KITTI 2015 | | | |
|---|---|---|---|---|---|---|---|
| | | | | Non Occlusion 3PE | | All Areas 3PE | |
| | EPE | 1PE | 3PE | Foreground | All | Foreground | All |
| MC-CNN [53] | 3.79 | - | - | 7.64 | 3.33 | 8.88 | 3.89 |
| GC-Net [12] | 2.51 | 16.9 | 9.34 | 5.58 | 2.61 | 6.16 | 2.87 |
| PSMNet [4] | 1.09 | 12.1 | 4.56 | 4.31 | 2.14 | 4.62 | 2.32 |
| SegStereo [49] | 1.45 | - | - | 3.70 | 2.08 | 4.07 | 2.25 |
| GwcNet-g [11] | 0.77 | 8.0 | 3.30 | 3.49 | 1.92 | 3.93 | 2.11 |
| HD$^3$-Stereo [50] | 1.08 | - | - | 3.43 | 1.87 | 3.63 | 2.02 |
| GANet [54] | 0.84 | 9.9 | - | 3.37 | 1.73 | 3.82 | 1.93 |
| AcfNet [55] | 0.87 | - | 4.31 | 3.49 | 1.72 | 3.80 | 1.89 |
| Stereo Expansion [48] | - | - | - | 3.11 | 1.63 | 3.46 | 1.81 |
| GANet Deep [54] | 0.78 | 8.7 | - | 3.11 | **1.63** | 3.46 | **1.81** |
| **CDN**-PSMNet | 0.98 | 9.1 | 3.99 | 4.01 | 2.12 | 4.34 | 2.29 |
| **CDN**-GANet Deep | **0.70** | **7.7** | **2.98** | **2.79** | 1.72 | **3.20** | 1.92 |

following [7, 44]. We report the average precision (AP) at IoU thresholds 0.5 and 0.7. We denote AP for the 3D and BEV tasks by $AP_{3D}$ and $AP_{BEV}$, respectively. The benchmark defines for each category three cases — easy, moderate, and hard – according to the bounding box height and occlusion and truncation. In general, the easy cases correspond to cars within 30 meters of the ego-car distance.

## 4.2 Implementation details

We mainly use the Wasserstein-1 distance (i.e., $W_1$ loss) for training our **CDN** model. We compare $W_1$ and $W_2^2$ losses in the supplementary material.

**Stereo disparity.** We apply our **continuous disparity network (CDN)** architecture to PSMNet [4] and GANet [54], namely **CDN**-PSMNET and **CDN**-GANET. To keep a fair comparison, we train the models with their default settings. For Scene Flow, the models are trained from scratch with a constant learning rate of 0.001 for 10 epochs. For KITTI 2015, the models pre-trained on Scene Flow are fine-tuned following the default strategy of the vanilla models. We consider disparities in the range of $[0, 191]$ for both datasets. We use a uniform grid of bin size 2 pixels to create the categorical distribution (cf. Equation 5). We show the effect of bin sizes in the supplementary material.

**Stereo depth.** We apply **CDN** to the SDN architecture [51], namely **CDN**-SDN. We follow the training procedure in [51]. We consider depths in the range of $[0m, 80m]$. We use a uniform grid of bin size 1m to create the categorical distribution.

**The offset sub-network.** We implement $b(u, v, d)$ with a Conv3D-Relu-Conv3D block. It takes the 4D cost volume, before the last fully-connected or convolutional block of $S_{disp}(u, v, d)$, as the input. We predict a single offset $b(u, v, d) \in [0, s]$ for each integral disparity value $d$, where $s$ is the bin size. We achieve this by clipping. The sub-network has 30K parameters, only $0.3\%$ w.r.t. PSMNET [4]. For stereo depth, we implement $b(u, v, z) \in [0, s]$ in the same way for each integral depth value $z$.

**Stereo 3D object detection.** We apply **CDN**-SDN to PSEUDO-LiDAR ++ [51], which uses SDN to estimate depth. We fine-tune the **CDN**-SDN model pre-trained on Scene Flow on KITTI 3D dataset, followed by using an 3D object detector, here P-RCNN [35], to detect 3D bounding boxes of cars. We also apply **CDN** to DSGN [8], the state-of-the-art stereo-based 3D object detector. DSGN uses as a backbone depth estimator based on PSMNET and we replace it with our **CDN** version.

**Multi-modal ground truths.** As mentioned in subsection 3.3, we create multi-modal ground truths for a pixel by considering a patch in its $k \times k$ neighborhood. We give the center-pixel disparity a weight $\alpha = 0.8$, and the remaining ones an equal weight such that the total sums to 1. In this case, we use Equation 12 as the loss function. We implement a differentiable loss module in Pytorch that can be applied to a batch of image tensors. Please see the supplementary material for more details.

Table 2: **3D object detection results on the KITTI leader board.** We report $AP_{BEV}$ and $AP_{3D}$ (in %) of the **car** category at IoU$=0.7$. Methods with **CDN** are in blue. The best result of each column is in bold font.

| Method | BEV Detection AP ($AP_{BEV}$) | | | 3D Detection AP ($AP_{3D}$) | | |
|---|---|---|---|---|---|---|
| | Easy | Moderate | Hard | Easy | Moderate | Hard |
| S-RCNN [19] | 61.9 | 41.3 | 33.4 | 47.6 | 30.2 | 23.7 |
| OC-STEREO [29] | 68.9 | 51.5 | 43.0 | 55.2 | 37.6 | 30.3 |
| DISP R-CNN [36] | 74.1 | 52.4 | 43.8 | 59.6 | 39.4 | 32.0 |
| PSEUDO-LIDAR [40] | 67.3 | 45.0 | 38.4 | 54.5 | 34.1 | 28.3 |
| PSEUDO-LIDAR ++ [51] | 78.3 | 58.0 | 51.3 | 61.1 | 42.4 | 37.0 |
| PSEUDO-LIDAR E2E [31] | 79.6 | 58.8 | 52.1 | 64.8 | 43.9 | 38.1 |
| **CDN**-PSEUDO-LIDAR ++ | 81.3 | 61.0 | 52.8 | 64.3 | 44.9 | 38.1 |
| DSGN [8] | 82.9 | 65.0 | 56.6 | 73.5 | 52.2 | 45.1 |
| **CDN**-DSGN | **83.3** | **66.2** | **57.7** | **74.5** | **54.2** | **46.4** |

Table 3: **Disparity multi-modal results.** We report the EPE, 1PE and 3PE on Scene Flow. Methods with **CDN** are highlighted in blue. The best result of each column is in bold font.

| Method | EPE | 1PE | 3PE | Method | EPE | 1PE | 3PE |
|---|---|---|---|---|---|---|---|
| PSMNET [4] | 1.09 | 12.1 | 4.56 | GANET Deep [54] | 0.78 | 8.7 | - |
| **CDN**-PSMNET | 0.98 | 9.1 | 3.99 | **CDN**-GANET Deep | 0.70 | 7.7 | 2.98 |
| **CDN**-PSMNET MM | **0.96** | **9.0** | **3.96** | **CDN**-GANET Deep MM | **0.68** | **7.7** | **2.97** |

## 4.3 Main results

**Disparity estimation.** Table 1 summarizes the results on disparity estimation. **CDN**-GANET Deep[3] achieves the lowest error at all three metrics on Scene Flow. It reduces the error for GANET Deep by 1.0 1PE and 0.08 EPE, both are significant. We see a similar gain for PSMNET: **CDN**-PSMNET reduces EPE by 0.09, demonstrating the general applicability of our approach to existing networks.

On KITTI 2015, **CDN**-GANET Deep obtains the lowest error on the *foreground* pixels and performs comparably to other methods on all the pixels[4]. We see a similar gain by **CDN**-PSMNET over PSM-NET on the foreground, which is quite surprising, as we do not specifically *re-weight* the loss function towards foreground pixels. Since **CDN** has advantages on pixels whose disparity is ambiguous and hard to estimate correctly (e.g., due to multi-modal distributions), the fact that foreground pixels have a higher error and **CDN** can effectively reduce it suggests that those challenging pixels are mostly in the foreground. As will be seen in 3D object detection, the improvement by **CDN** on foreground pixels translates to a higher accuracy on localizing objects.

**3D object detection.** Table 2 summarizes the results on the test set of KITTI 3D. Our **CDN** consistently improves the two mainstream approaches, namely, DSGN and PSEUDO-LIDAR. For PSEUDO-LIDAR, we achieve a 2.5%/3.0% gain on $AP_{3D}/AP_{BEV}$ Moderate (the standard metric on the leader board) against PSEUDO-LIDAR ++: the only difference is that we replace SDN by our **CDN**-SDN to have better depth estimates. Our approach even outperforms PSEUDO-LIDAR E2E, which fine-tunes the depth network specifically for object detection. We argue that our approach, which can *automatically* focus on the foregrounds, may have a similar effect as end-to-end training with object detection losses. For DSGN, plugging our **CDN**-SDN leads to a notable 2% gain at $AP_{3D}$, attaining the highest entry of stereo-based 3D detection accuracy on the KITTI leader board.

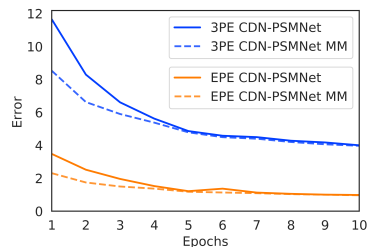

Figure 4: **MM training.** We show the EPE and 3PE disparity errors on Scene Flow test set using **CDN**-PSMNET, w/ or w/o MM training. MM training leads to faster convergence.

Table 4: **Depth multi-modal results.** We report the RMSE and ABSR errors on Scene Flow. The best result of each column is in bold font.

| Method | RMSE (m) | ABSR |
|---|---|---|
| SDN [51] | 2.05 | 0.039 |
| **CDN**-SDN | 1.81 | 0.030 |
| **CDN**-SDN MM | **1.80** | **0.028** |

Table 5: **Ambiguous regions (object boundaries).** We report the disparity error on Scene Flow. The best result of each column is in bold font.

| Method | EPE | 1PE | 3PE |
|---|---|---|---|
| PSMNet [4] | 3.10 | 20.1 | 11.33 |
| CDN-PSMNet | 2.10 | 15.3 | 8.92 |
| CDN-PSMNet MM | **2.08** | **13.2** | **8.65** |

### 4.4 Analysis

**Multi-modal (MM) ground truth.** We investigate creating the multi-modal (MM) ground truths for training our models. Table 3 and Table 4 summarize the results on Scene Flow for disparity and depth estimation, respectively. MM training slightly reduces the errors. To better understand how MM ground truths affect network training, we plot the test accuracy along the training epochs in Figure 4: **CDN**-PSMNET trained with MM ground truths converges much faster. We attribute this to the observations in [1]: a neural network tends to learn simple and *clean* patterns first. We note that, for boundary pixels whose disparities are inherently multi-modal, uni-modal ground truths are indeed *noisy* labels. A network thus tends to ignore these pixels in the early epochs. In contrast, MM ground truths provide *clean* supervisions for these boundary pixels; the network thus can learn the patterns much faster. See the supplementary material for a visualization and further discussions.

**Ablation studies.** We study different components of our approach in Table 6. Methods without $W_1$ loss use the regression loss for optimization (cf. Equation 2) and output the mean. Methods with $W_1$ loss output the mode. We see that, the offset sub-network alone can hardly improve the performance. Using $W_1$ distance alone reduces 1PE and 3PE errors, but not EPE, suggesting that it cannot produce sub-pixel disparity estimates[5]. Only combining the offset sub-network and the $W_1$ loss produces consistent improvement over all three metrics.

Table 6: **Ablation studies.** We report disparity error for **CDN**-PSMNET on Scene Flow. Methods without $W_1$ loss are learned with *mean* regression.

| Offsets | $W_1$ Loss | Output | EPE | 1PE | 3PE |
|---|---|---|---|---|---|
| | | Mean | 1.09 | 12.1 | 4.56 |
| ✓ | | Mean | 1.04 | 12.0 | 4.55 |
| | ✓ | Mode | 1.20 | 10.5 | 4.21 |
| ✓ | ✓ | Mode | **0.98** | **9.1** | **3.99** |

**Disparity on boundaries.** Table 5 shows the results: we obtain pixels on object boundaries using the OpenCV Canny edge detector with minVal/maxVal=100/200. Both **CDN** and training with multi-modal ground truths reduce the error significantly.

**Qualitative disparity results on KITTI.** As shown in Figure 5, our approach is able to estimate disparity accurately, especially along the object boundaries. Specifically, **CDN**-GANET Deep maintains the straight bar shape (on the right), while GANET Deep blends it with the background sky due to the mean estimates.

## 5 Conclusion

In this paper we have introduced a new output representation, model architecture and loss function for depth/disparity estimation that can faithfully produce real-valued estimates of depth/disparity. We have shown that results not just in more accurate depth estimates, but also significant improvement in downstream tasks like object detection. Finally, because we explicitly output and optimize a distribution over depths, our approach can naturally take into account *uncertainty and multimodality* in the ground truth. More generally, our results suggest that removing suboptimalities in how we represent and optimize 3D information can have a large impact on a multitude of vision tasks.

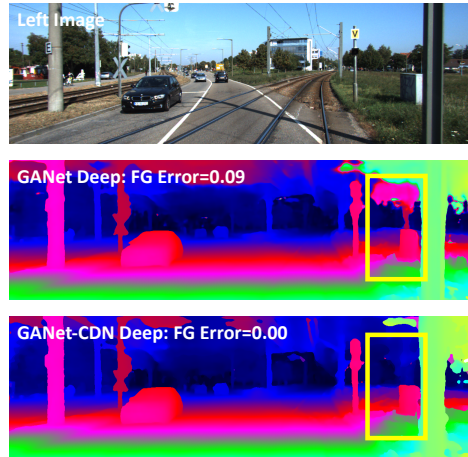

Figure 5: **Qualitative results on disparity.** The top, middle, and bottom images are the left image, the result of GANET Deep, and the result of **CDN**-GANET Deep, together with the foreground 3PE.

## Broader Impact

The end results of this paper are improved depth and disparity estimation, particularly on foreground objects. This is of use to self-driving cars, 3D reconstruction, and other robotics applications. In particular, it has the potential to improve the *safety* of these systems, as indicated by the increased 3D object detection performance. Our approach can also easily be incorporated into other depth or disparity estimation algorithms for further improvement.

While our depth predictions are significantly better, any failure has important safety considerations, such as collisions and accidents. Before deployment, appropriate safety thresholds must be cleared.

Our approach does not specifically leverage dataset biases, although being a machine learning approach, it is impacted as much as other machine learning techniques.

## Acknowledgments

This research is supported by grants from the National Science Foundation NSF (III-1618134, III-1526012, IIS-1149882, IIS-1724282, and TRIPODS-1740822, OAC-1934714), the Office of Naval Research DOD (N00014-17-1-2175), the Bill and Melinda Gates Foundation, and the Cornell Center for Materials Research with funding from the NSF MRSEC program (DMR-1719875). We are thankful for generous support by Zillow, SAP America Inc, AWS Cloud Credits for Research, Ohio Supercomputer Center, and Facebook.

## Footnotes

[1]Our work is reminiscent of G-RMI pose estimator [28], which predicts the heatmaps (at fixed locations) and offsets for each keypoint. Our work is also related to one-stage object detectors [20, 22, 33] that predict the class probabilities and box offsets for each anchor box.

[2]For dealing with disparity or depth values at a pixel, our metric space naturally becomes $\mathcal{R}^1$.

[3]We apply the GANET Deep model introduced in the released code of [54], available at https://github.com/feihuzhang/GANet. The main architectures of GANET Deep and GANET are the same, while the former has some more 2D and 3D convolutional layers.

[4]There are two possible reasons that **CDN**-GANET Deep does not outperform GANET Deep on all the pixels. First, **CDN** overly focuses on foreground pixels. Second, we used the same hyper-parameters as the original GANET without specific tuning for **CDN**. We note that the ratio of foreground/background pixels is $\sim 0.15/0.85$; the degradation by **CDN** on the background is $\sim 0.16$ 3PE, smaller than the gain on foreground.

[5]Using a bin size $s = 2$ without offsets, the mode is restricted to integral values and EPE suffers.

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
