[Supplementary Material]

# Supplementary Material:
# Wasserstein Distances for Stereo Disparity Estimation

**Divyansh Garg**[1]  **Yan Wang**[1]  **Bharath Hariharan**[1]
**Mark Campbell**[1]  **Kilian Q. Weinberger**[1]  **Wei-Lun Chao**[2]
[1]Cornell University, Ithaca, NY    [2]The Ohio State University, Columbus, OH
{dg595, yw763, bh497, mc288, kqw4}@cornell.edu    chao.209@osu.edu

We provide in this material the contents omitted in the main paper:

- section 1: additional implementation details (cf. subsection 3.2 and subsection 4.2 of the main paper).

- section 2: additional discussions (cf. section 3 and subsection 4.4 of the main paper).

- section 3: additional experimental results and analysis (cf. subsection 4.2, subsection 4.3, and subsection 4.4 of the main paper).

## 1 Implementation Details

### 1.1 Learning with multi-modal ground truths

For multi-modal ground truths, we cannot use Equation 8 of the main paper for optimization. Instead, we apply the loss in Equation 12, for $W_1$ distance. This loss essentially computes the difference in areas between the CDFs of the two distributions. For mixtures of Dirac delta functions, it can be efficiently implemented by computing the accumulated difference between CDF histograms. It takes $\mathcal{O}(B \log B)$ for each pixel using sorting, where $B$ is the total number of supports of both distributions. Our implementation is adapted from *scipy.stats.wasserstein_distance* and we modify it to be compatible with Pytorch tensors and use CUDA to parallelize the computation over all the pixels.

### 1.2 Learning with the (approximated) KL divergence

The *Kullback–Leibler* (KL) divergence

$$KL(\mu||\nu) = \mathbb{E}_\mu \log(\mu|\nu) \tag{1}$$

between two distributions $\mu$ and $\nu$ requires them to have the same supports: i.e., $\mu(d') = 0$ if $\nu(d') = 0$, for $KL(\mu||\nu)$ to be finite.

For our case, $\mu(d') = \delta(d' - d^\star)$ and $\nu(d') = \sum_{d \in \mathcal{D}} p(d|u,v)\delta(d' - (d + b(u,v,d)))$. These two measures may have different supports. To make the KL divergence applicable, we can smooth $\nu$ to form a mixture of Laplace or Gaussian distributions.

For example, smoothing $\nu$ with a Laplace distribution, $\text{Laplace}(0,\tau) = \frac{1}{2\tau} \exp\left(-\frac{|d'|}{\tau}\right)$, we get

$$\nu_{\text{Lap}}(d') = \sum_{d \in \mathcal{D}} p(d|u,v) \frac{1}{2\tau} \exp\left(-\frac{|d' - (d + b(u,v,d))|}{\tau}\right). \tag{2}$$

With $\nu_{\text{Lap}}$, the KL divergence reduces to the following loss

$$\ell(\mu, \nu_{\text{Lap}}) = -\log \sum_{d \in \mathcal{D}} p(d|u,v) \frac{1}{2\tau} \exp\left(-\frac{|d^\star - (d + b(u,v,d))|}{\tau}\right) \tag{3}$$

$$\approx -\log p(\bar{d}|u,v) + \frac{1}{\tau}\left|d^\star - (\bar{d} + b(u,v,\bar{d}))\right|, \tag{4}$$

where $\bar{d} = s\lfloor\frac{d^\star}{s}\rfloor \in \mathcal{D}$ is the grid disparity value of the bin the true disparity $d^\star$ belongs to.

Similarly, smoothing $\nu$ with a Gaussian distribution $\mathcal{N}(0, \sigma^2)$, we get

$$\nu_{\text{Gau}}(d') = \sum_{d \in \mathcal{D}} p(d|u,v) \frac{1}{\sigma\sqrt{2\pi}} \exp\left(-\frac{\|d' - (d + b(u,v,d))\|_2^2}{2\sigma^2}\right). \tag{5}$$

With $\nu_{\text{Gau}}$, the KL divergence reduces to the following loss

$$\ell(\mu, \nu_{\text{Gau}}) = -\log \sum_{d \in \mathcal{D}} p(d|u,v) \frac{1}{\sigma\sqrt{2\pi}} \exp\left(-\frac{\|d^\star - (d + b(u,v,d))\|_2^2}{2\sigma^2}\right) \tag{6}$$

$$\approx -\log p(\bar{d}|u,v) + \frac{1}{2\sigma^2}\left\|d^\star - (\bar{d} + b(u,v,\bar{d}))\right\|_2^2, \tag{7}$$

where $\bar{d} = s\lfloor\frac{d^\star}{s}\rfloor \in \mathcal{D}$ is the grid disparity value of the bin the true disparity $d^\star$ belongs to.

These formulations reduce to the conventional classification loss plus offset regression loss, commonly used for keypoint estimation [5, 9] and one-stage 2D object detection [3, 4, 6, 9].

## 2 Additional Discussions

### 2.1 Multi-modal ground truths

There are three reasons why multi-modal ground truths would benefit disparity or depth estimation. First, pixels are discrete: a single pixel may capture different depths. Second, real datasets need to project signals from a depth sensor (e.g., LiDAR) to a depth map. As pixels are discrete and the cameras and LiDAR might be placed differently, multiple LiDAR points of different depths may be projected to the same pixel. Third, for stereo estimation, pixels along boundaries or occluded regions cause ambiguity to the model; multi-modal ground truths offer better supervision for training, especially in early training epochs.

Conceptually, learning with multi-modal ground truths should notably improve results in Table 3 and Table 4 of the main paper. However, in evaluation, a majority of pixels are not on the object boundaries. Besides, we still evaluate using the (likely noisy) uni-modal ground truths. To further analyze these, we show in Table 6 of the main paper the disparity error calculated on object boundaries: learning with multi-modal ground truths leads to a significant improvement.

### 2.2 Multi-task learning

One way to mitigate stereo predictions at depth discontinuities is to jointly perform stereo estimation and other tasks such as semantic segmentation [1, 2, 7], which can reason about object boundaries. The core idea is to leverage additional semantic labels to guide the model to resolve depth discontinuities (i.e., predict uni-modal distributions). Our method, in contrast, does not prevent predicting multi-modal distributions along depth discontinuities, but changes the outputting rule (i.e., argmin with a predicted offset). Our method can also capture depth discontinuities within an object or an object class. In contrast, semantic segmentation labels overlapped objects of the same class by the same label and does not directly tell their boundaries.

### 2.3 Offsets and distributions without common supports

While the *learned* offsets may lead to common supports between the predicted and ground truth distributions, we have to first come up with a loss to *learn* such offsets before common supports become possible. Concretely, to learn $b(u,v,d)$ in Equation 9, we need a loss that can measure the

Table 1: **Comparison of different divergences (distances).** We report the RMSE and the ABSR error for depth estimation on Scene Flow. The best result of each column is in bold font.

| Method | Divergence | RMSE (m) | ABSR |
|--------|-----------|----------|------|
| SDN | - | 2.05 | 0.04 |
| **CDN**-SDN | KL | 2.57 | 0.04 |
| **CDN**-SDN | $W_1$ | **1.81** | **0.03** |
| **CDN**-SDN | $W_2^2$ | 1.91 | 0.05 |

Table 2: **Comparison of bin sizes.** We report the disparity error on Scene Flow using **CDN**-PSMNET model.

| Bin size | EPE | 1PE | 3PE |
|----------|-----|-----|-----|
| 1 | 1.22 | 13.9 | 4.33 |
| 2 | **0.98** | **9.1** | **3.99** |
| 4 | 1.52 | 26.1 | 4.17 |

divergence between $\tilde{p}$ and $p^\star$, which may not have common supports. We note that, this may occur even if the target distribution $p^\star$ is a Dirac delta function. While the KL divergence or a regression loss may be applied to learn the offsets, they need to either smooth the distributions or carefully design the loss to learn both the distribution and the offset networks. The Wasserstein distance offers a principled loss to learn the two networks jointly.

## 3 Additional Results and Analysis

### 3.1 Ablation studies on different divergences

We show the ablation study on using different divergences between distributions in Table 1. For the KL divergence (subsection 1.2), we use Laplace smoothing with $b = 1$ (Equation 4). Our results show that the Wasserstein distance is a better choice than the KL divergence for comparing the predicted and the ground truth disparity (or depth) distributions. We also see that $W_2^2$ distance performs worse than $W_1$. We attribute this to outliers (i.e., noisy disparity labels) in a dataset.

### 3.2 Effect of bin sizes

**CDN** outputs modes and needs (a) the bin containing the truth disparity or depth to have the highest probability and (b) the offset to be accurate. The bin size balances the difficulty of (a) and (b). A smaller bin size makes (a) harder. A larger bin size makes (a) easier but makes (b) harder as the range of offsets gets larger. It is the only hyper-parameter to tune and only integral values are considered.

We show these effects of bin sizes on uniform grids, with disparities in the range of $[0, 191]$ for disparity estimation in Table 2. For a bin size $s = 1$, predicting the correct bin is harder. For a bin size $s = 4$, predicting the correct bin is easier, whereas predicting the correct offset becomes harder. We found $s = 2$ to perform well in general.

### 3.3 Ablation studies on $\alpha$ and $k$ in multi-modal (MM) ground truths

Table 3 shows the depth estimation error on Scene Flow using **CDN**-SDN-MM, with different $\alpha$ and $k$ in preparing the MM ground truths (cf. Table 4 of the main paper). A smaller $\alpha$ leads to a larger error, which makes sense as it relies less on the ground truths.

### 3.4 Learning with multi-modal (MM) ground truths

Following subsection 4.4 and Figure 4 of the main paper, we train **CDN**-PSMNET and **CDN**-PSMNET MM for only two epochs and compare their disparity estimation performance. Figure 1 shows the results on KITTI images. While both methods have similar predictions at smooth regions, **CDN**-PSMNET MM leads to much sharper and clearer object boundaries, suggesting that the multi-modal ground truths are better supervisions for learning around the boundaries in early epochs.

Table 3: **Ablation studies on the MM ground truths.** We conduct experiments using **CDN**-SDN MM on Scene Flow (cf. Table 4 of the main paper).

| $\alpha$ | k | RMSE | ABSR |
|---|---|---|---|
| 0.8 | 3 | **1.80** | **0.028** |
| 0.8 | 5 | 1.82 | 0.029 |
| 0.8 | 7 | 1.88 | 0.035 |
| 0.5 | 3 | 1.81 | 0.029 |
| 0.2 | 3 | 2.20 | 0.062 |

Figure 1: **Qualitative results on learning with multi-modal ground truths at early epochs**. The top, middle, and bottom images are the left image, the result of **CDN**-PSMNET, and the result of **CDN**-PSMNET MM.

### 3.5 Learned offsets

The offset network learns to produce the sub-grid disparity at each integral disparity values. Figure 2 shows an example, in which we back-project pixels into 3D points using the estimated disparity or depth at each pixel by the *mode*, with or without the offset prediction. Without the offset, the 3D points can only occupy discrete depths, leading to a discontinuous, non-smooth point cloud.

### 3.6 Point cloud visualization

Figure 3 shows the BEV point cloud visualization. We show the 3D points generated by SDN and **CDN**-SDN as well as the ground truth LiDAR points and car/pedestrian boxes. We see that, **CDN**-SDN generates sharper points than SDN. Specifically for pixels on the foreground objects, SDN usually predicts the depths beyond the boxes due to the *mean* estimates from multi-modal distributions on the boundary pixels, whereas **CDN**-SDN significantly alleviates the problem. We also see some failure cases of **CDN**-SDN: on the right image, **CDN**-SDN has a larger error on the background compared to SDN.

### 3.7 Depth estimation

Besides the Scene Flow dataset, we show the depth estimation error on KITTI Val: the 3,769 validation images for 3D object detection. We follow [8] to train the depth estimation model and compute the depth estimation error on pixels associated with ground truth LiDAR points. Table 4 and Figure 4 show the results, **CDN**-SDN achieves lower error than SDN, which explains why **CDN**-SDN (and **CDN**-DSGN) can lead to better 3D object detection accuracy.

Figure 2: A visualization of the 3D point cloud (from the bird's-eye view) derived from the estimated disparities or depths (by modes), with (right) and without (left) offset prediction.

Figure 3: **BEV Point cloud visualization.** The blue points are obtained using SDN. The red points are from our **CDN**-SDN model. The yellow points are from the ground truth LiDAR. The green boxes are ground truth car / pedestrian locations. The observer is at the left-hand side of the point cloud looking to the right.

Table 4: **Depth error on KITTI Val.** We compare SDN and **CDN**-SDN models.

| Method | Depth errors (m) | | | |
|---|---|---|---|---|
| | Mean | Median | RMSE | ABSR |
| SDN | 0.589 | 0.128 | 3.08 | 0.044 |
| **CDN**-SDN | **0.524** | **0.093** | **3.00** | **0.042** |

Figure 4: **Depth error on KITTI Val.** We compute the median absolute depth error for different depth ranges on KITTI Val images using SDN and **CDN**-SDN.

Figure 5: We show the object detection precision-recall curves for $AP_{3D}$ at moderate cases on cars. We compare DSGN (stereo images) and **CDN**-DSGN (stereo images).

### 3.8 3D object detection

We show in Figure 5 the object detection precision-recall curves of DSGN vs. **CDN**-DSGN. **CDN**-DSGN has higher precision (vertical) values than DSGN at different recall (horizontal) values.

### 3.9 Qualitative disparity results

We show in Figure 6 and Figure 7 the predicted disparity maps and the foreground errors of both GANET Deep and **CDN**-GANET Deep on KITTI and Scene Flow. **CDN** generally leads to sharper and clearer object boundaries.