[Reviews · NeurIPS 2020]

Review 1

Summary and Contributions: In this paper, the authors tackle the stereo matching problem by first analyzing the main weaknesses of existing deep architectures to represent continuous depth estimates and subsequently proposing effective changes to address them. More specifically, after highlighting how such techniques, typically based on the computation of the mean of a disparity distribution, can cause wrong depth predictions when the estimated distribution is multi-modal (e.g. near object boundaries), they introduce 1) a novel output representation built on a sub-network that predicts offset values aimed at computing distribution over arbitrary disparities from which it is possible to predict the continuous mode 2) a novel loss function based on the Wasserstein metric to minimize the distance between the predicted distribution and the real-one. Moreover, by taking advantage of distribution matching, they also propose to exploit multi-modal ground-truth at each pixel location. Experimental results on the real-world KITTI 2015 dataset and the synthetic SceneFlow dataset show the effectiveness of the proposed strategy applied to different 3D stereo architectures (suited for disparity or depth estimation). In addition, the reported ablation study demonstrates how the contributions progressively improve depth and allow (in specific cases) for faster convergence. Finally, the authors show how the 3D object detection task can also benefit from this approach in autonomous driving scenarios, reaching state-of-the-art results.

Strengths: - The authors carefully motivated the proposal, analyzing in detail the impact of the depth predicted by existing deep stereo networks on real-world applications. They clearly show with a practical example that wrong depth values can have negative effects on subsequent tasks such as 3D object detection and, for this reason, they propose to identify the main problem associated with this and propose a solution for it. - It is to be welcomed that the related work contains a short but detailed mathematical description of how continuous depth is computed by existing strategies that construct a cost volume over an integer-valued disparity dimension. This facilitates reading and understanding of subsequent sections. - The method description (Section 3) is rigorous and detailed. - In general, the contributions of the paper are simple and easy to reproduce. This methodology can be also deployed by other stereo architectures based on the same architectural principles of PSMNet, GANet and SDN (chosen for experiments in this work). For this reason, the use of different architectures in order to validate the proposal is an added value. - The supplementary material provides further experiments and insights about the proposed technique. The point cloud visualization (Figure S1) clearly show how this strategy notably improves depth near object boundaries compared to the state-of-the art SDN network. Moreover, the depth error analysis (Figure S2) on KITTI Val allows to better understand the impact at different depth ranges. - The reported ablation study on the SceneFlow dataset shows the importance of each contribution in the final depth/disparity prediction for SDN and PSMNet architectures. - Experimental results on real-world applications such as 3D object detection on KITTI are of considerable interest that further demonstrate the importance of the proposal.

Weaknesses: - An ablation study on the GANet architecture is missing (also in the supplementary material). It would be interesting to analyse the effect of each contribution also on this network, which currently represents the state-of-the-art on the disparity estimation task. Moreover, the effectiveness (also in terms of convergence) of multi-modal learning is not demonstrated on it. - Section 3.3 (Extension: learning with multi-modal ground-truth). The concept of multiple ground-truth depth/disparity values is not so clear. For example, why should a synthetic dataset like SceneFlow provide different ground-truth values for each pixel? Is it something suited for realistic datasets (e.g. KITTI) in which ground-truth depth values could be estimated from different sensors (combination of passive and active sensors)? I would like to have an in-depth analysis of it. - Table 1 seems to report results without the MM learning. This is not well specified in the paper, but by comparing CDN-PSMNet'results of Table 3 with Table 1 it would seem to be so. If that's the case, why didn't the authors evaluate their final model (CDN-PSMNet MM/CDN-GANet Deep MM) in Table 1 for each dataset? - Table 3 and Table 4 report disparity and depth multi-modal results (MM) on PSMNet and SDN, respectively. However, by looking at the numbers, they look like fluctuations due to training, as there seem to be no particular advantages in terms of depth errors. In section 4.3 the authors state that "We note that, for boundary pixels whose disparities are inherently multi-modal, uni-modal ground-truths are indeed noisy labels". This mean that, in addition to a faster convergence, MM should provide much stronger supervision, especially at depth discontinuities but this is not proven. I suggest performing the "4.3 Disparity on boundaries" analysis with and without multi-modal ground-truths in order to demonstrate this fact and verify if there is an improvement in such regions. - The importance of the \alpha parameter described in Section 4.2 (Multi-modal (MM) ground-truth) is not clear. The ablation study on multi-modal learning does not show particular advantages. The \alpha value could be a key element but there is no evidence about this. The same concept could be applied to the k parameter. - Section 4.3 (Disparity on boundaries). The evaluation protocol for boundaries is not clear from the paper. A better explanation of the evaluation protocol is needed. Moreover, information regarding the Canny Edge Detector is also missing in the supplementary material (e.g. minVal, maxVal) - Section S1.2 poorly describes the offset sub-network. Moreover, further details like number of added parameters, the adopted activation function and the impact on efficiency/memory requirements are missing. - Section S2.2 (Effect of bin sizes). The authors did not give a detailed and convincing explanation of the effect of bin sizes, despite the ablation reported in Table S2. A further study is recommended.

Correctness: I believe that both the claim and the method are correct and well motivated. Empirical demonstrations are also conducted correctly, following typical practices used in the stereo matching field.

Clarity: The paper is easy to follow and quality of writing is, on the whole, very good.

Relation to Prior Work: The authors described in detail the main differences from previous contributions, highlighting the most similar existing works (line 108) and their different peculiarities.

Reproducibility: Yes

Additional Feedback: - By reading the work of Kendall et al. [12] (Section 3.4 - Differentiable ArgMin) they show that, to overcome multi-modal distributions, the network's regularization based on 3D convolutions produces a disparity probability distribution which is predominantly unimodal, thus solving the wrong estimation based on the mere mean. Could the authors briefly comment on this by referring to Figure 2 of their paper? - In section 4.2/4.3 the difference between GANet Deep [46] and GANet [46] (Table 1) is not clear. Please, briefly describe the difference. - Could the authors explain why CDN-GANet Deep is slightly worse than the original GANet Deep network in "All Areas" on the KITTI 2015 dataset despite the improvement on foreground regions? - I suggest adding an in-depth description of training details in the supplementary material (e.g. image/crop resolution, epochs, batch size, optimizer, hardware used for experiments and so on). - I suggest adding more qualitative results in the supplementary material (e.g., depth/disparity map + error map), especially for the SceneFlow dataset in which it is possible to visually perceive improvements near depth boundaries. Moreover, I believe that Figure 4 in the main paper should be enriched with the corresponding error maps or, at least, error metrics (3PE) in the caption. - References to competitors are missing from Table 2/3/4/6 of the main paper as well as from Table S1/S2/S3 of the supplementary material. - Some pivotal references for stereo are missing, for example: ** D. Scharstein and R. Szeliski. A taxonomy and evaluation of dense two-frame stereo correspondence algorithms. International journal of computer vision, 47(1-3):7–42, 2002. - (minor) Line 241: "We use a uniform grid of bin size 2" -> "We use a uniform grid of bin size s=2" - (minor) If I may say, I believe the title can be improved. By reading "Wasserstein Distances for Stereo Disparity Estimation", it might seem that it is a methodology suited for estimating disparity only despite the authors demonstrate that it is also possible to apply it on the depth domain directly. Moreover, the strategy is not just limited to Wasserstein distances but to a broader class of divergence measures. Update after rebuttal: After reading the author's rebuttal and comments of other reviewers, I still think positively about this paper. We all agree that it is well-motivated, easy to reproduce and effective. Moreover, some concerns raised during the first phase of reviews have been clearly addressed in the rebuttal, in which some crucial aspects have been deepened. More specifically, I found clarifications about the multi-modal ground-truth, the KL-divergence (raised by R2) and comparison with [12] reasonable and sound. Also, additional quantitative results reported in the rebuttal (missing in the paper) help to better understand the impact of important hyper-parameters, thus confirming the effectiveness of the proposed strategy. I would strongly suggest the authors to better clarify all these aspects in the main paper and enrich the supplementary material accordingly. Overall, I believe that the proposed strategy helps to alleviate an existing problem in the stereo matching task using a novel perspective and, although there isn't a huge gap with respect other state-of-the-art on standard disparity metrics, it is clear how this allows to achieve better results on specific regions (depth discontinuities/foreground regions) and on downstream tasks such as 3D object detection. Therefore, I believe that the rebuttal phase has allowed to clarify some controversial aspects and that the paper deserves to be accepted.


Review 2

Summary and Contributions: This paper proposed a novel stereo estimation network. The core contribution is a new header that predicts a small continuous offset in disparity, and the usage of Wasserstein motivated loss function. The key idea underlying is to better model the multi-modal continuous distribution over the disparity space. The author experimented on stereo matching as well as stereo based object detection. The approach achieved impressive results bringing improvement over multiple tasks / multiple backbones. It’s interesting to see the proposed method has a ubiquitous performance improvement. And I am quite convinced about the formulation in Wasserstein distance between two mixtures of Dirac. Yet it’s not clear why the usage of a small continuous sub-disparity offset could contribute to the success.

Strengths: - Motivated clear - AFAIK, contribution is novel in the domain of stereo matching - Presentation is clear. Paper is well-written. Mathematical notations are clear. - Experiments show improvements over multiple methods and tasks.

Weaknesses: There are several aspects that I think the author could improve on. 1. The author could provide better insight on what has been learned for the offset, and provide qualitative examples; 2. The author should conduct a more thorough literature study on the deep stereo methods, as a plethora of important papers is missing. KL-divergence perspective: When there is no multi-modal GT, I found it’s hard to justify the usage of Wasserstein distance, as target distribution is simply a Dirac. S 1.3. provides a helpful derivation of Eq. 10 based on KL divergence in which the prediction is a complicated mixture of Gaussian/Laplacians. However, I am wondering if you could explain this from a different perspective. That is to say, Eq. 10 could be interpreted in a way that we are minimizing the cross-entropy / KL divergence between prediction distribution p and target distribution p* over discrete support space. But unlike the conventional method, in which p* is a predefined distribution, e.g., one-hot or soft-logit label computed from l2, the proposed method learns an offset to produce an adaptive target logit. I am wondering if the author could comment on this. Explanation/Insights: It’s good to see that adding a small continuous offset over the output space would bring a difference. However, I am wondering if the author has any insight into what the offset learns. E.g. sub-pixel disparity, better multi-modal support towards GT? I would strongly suggest the author could visualize the offset and offers some discussions. Moreover, the offset bin seems too small for me, which may suggest it merely learns a sub-pixel refinement. And it seems from Supp the well-tuned hyper-parameter choice is critical to achieving performance gain. Could you better justify this? Ablation on mode/mean: it’s quite surprising that the Wasserstein distance itself without offset brings such a significant drop in EPE. Do you use the mode or the mean in this ablation study for Wasserstein only? Have you tried both? Furthermore, have you tried to use PSMNet and use its mode for prediction instead of mean? What is the performance if you use mean as your final disparity estimation. Uncertainty: One thing interesting to me is it’s learned model uncertainty, which I expect to be better calibrated using Wasserstein distance than the soft-max logits trained with loss over soft-argmax mean. I suggest the author conduct a visualization of the learned mixture of Dirac PDF over some pixels, and a visualization of the dense uncertainty map (e.g. per pixel variance of the prediction, etc.). Example in Fig.4 does not explain the improvement and observations on the foreground. Maybe you could choose another example to showcase how foreground pixels have improved. Marginal improvement: The proposed method achieves a relatively marginal increase in terms of stereo estimation metrics. In this work, additional computation is brought by the offset header. However, as far as I know, at least for PSM-Net, a very light-weight disparity refinement header could easily bring additional improvement at the same level.

Correctness: Yes.

Clarity: Yes. See above.

Relation to Prior Work: Some important literature on deep stereo is missing. E.g. Žbontar & LeCun JMLR 16, Luo et al. CVPR 16, Kendall et al, CVPR 17. Some of which are relevant to the approach, e.g. discussions on the usage of mode vs mean, KL-divergence minimization for stereo, etc.

Reproducibility: Yes

Additional Feedback: ----------------- Post-rebuttall ---------------------------- The rebuttal addressed most of my concerns and I believe this paper develops a novel approach and refreshing perspective to push learning based stereo matching. The author did a great job in addressing the comments and clarify some unclear parts. The additional experiments are convincing. Thus I would recommend to accept this paper.


Review 3

Summary and Contributions: The authors pointed out two fundamental issues in current stereo estimation frameworks: (i) current systems first estimate a distribution over a fixed set of disparities, then compute the expected mean to be the final output. The average operation may result in inaccurate estimations that does not match the ground truth, especially for ambiguous regions. (ii) The regression loss cannot guarantee the model to learn the real distribution over the disparities, but only the final expected value. To overcome these limitations, the authors propose two adaptions to existing frameworks: (i) adding a branch to predict an offset for each disparity so that the model can output a distribution for arbitrary disparities; (ii) introducing a novel stereo loss based on Wasserstein distance to directly capture the uni-/multi-modal GT distribution. These two modifications can be easily plugged into the current systems. With them, the authors are able to greatly reduce the ambiguities in current stereo systems and achieve state-of-the-art results in 3D object detection.

Strengths: - the paper is very well-written. the authors first motivate the problem both intuitively and mathematically, then they propose a solution for it and provides insight into their method. the flow is very fluent and easy to follow. I enjoy reading it. - the proposed method is simple yet effective. It can be plugged into current mainstream models with very minor modifications. - while current dataset does not support multi-modal ground truth for stereo estimation, I like the idea of bringing this issue up and conducting some preliminary experiments. Interestingly, training with multi-modal signal seems to help in some cases (seems to be helpful on Sceneflow, but not sure about KITTI as the authors did not report that)

Weaknesses: - how does the CDN-SDN perform on KITTI stereo benchmark? The authors somehow only report its performance on KITTI 3D detection and Sceneflow and I wonder why? - Whats the rationale that "CDN-GANet Deep" performs worse than its original counterpart for "all"? If the foreground is better, then it's essentially implying that the background is doing way worse. Why is that? Can the authors provide more analysis on this? Also, I notice that the improvement on occluded regions is quite limited comparing to non-occluded regions. Does this imply that the newly proposed solution will harm the hallucination capability of the model?

Correctness: - I'm not sure if I agree the argument in L272-276. - the claim in L299-300 needs to be justified with evidence. How do the authors know that boundary pixels are learned later? Did they visualize it or measure it? If yes, can the report what you did?

Clarity: - I personally find it hard to parse Fig. 1. Even after reading the captions, I still cannot recognize the wall. I'd suggest the authors to replace that with some other more obvious examples. To my knowledge, its very easy to find such smearing effect on the cars in KITTI. Also, if you view it with right angle, people can see both the issue and the shape of the car...

Relation to Prior Work: yes

Reproducibility: Yes

Additional Feedback:


Review 4

Summary and Contributions: Current state of the art stereo estimation approaches compute depth as the expected valued over a distribution of discrete set of disparity values. However, this does not work well for object boundaries where the distribution is multi-modal due to depth discontinuities. The approach suggested by the paper handles this problem by two ways, i) Having an offset prediction branch to predict real disparity/depth values without taking an expectation, ii) Wasserstein based loss function which can compute divergence of multi-modal predictions and ground truths which do not necessarily share the same supports. This approach gives state of the art depth estimation results on standard benchmarks such as KITTI and Scene Flow. It also shows better performance in 3D object detection methods which use depth values from stereo estimation as part of its input.

Strengths: - There are two key contributions of this paper to predict better depth/disparity values at object boundaries. The first is the offset modeling which allows direct prediction of real valued depth and disparity values without needing to compute an expectation over the distribution. The second is coming up with a novel loss formulation which can compute divergence of the prediction and GT distributions which might not have the same support. - The paper is well written, clearly stating the novelty, deriving the loss and showing the effectiveness of its approach with well designed experiments.

Weaknesses: There are minor concerns, which if addressed would make the paper stronger (in my opinion). - Stereo estimation approaches have long suffered from the issues stated in this paper, i.e., reliability of stereo predictions at depth discontinuities. One of the ways to mitigate this is to jointly reason about stereo estimation as well as semantic segmentation, which can reason about object boundaries. This topic has been explored in papers such as a) Analyzing Modular CNN Architectures for Joint Depth Prediction and Semantic Segmentation by Jafari et. al. (ICRA '17) b) PAD-Net: Multi-Tasks Guided Prediction-and-Distillation Network for Simultaneous Depth Estimation and Scene Parsing by Xu et. al. (CVPR '18) c) Look Deeper into Depth: Monocular Depth Estimation with Semantic Booster and Attention-Driven Loss by Jiao et. al. (ECCV '18) While one of the key differences of this approach compared to the above is that this approach does not need additional semantic labeling GT. However, it would be interesting to demonstrate more differences (such as in terms of error modes) of these two kinds of approaches at handling stereo predictions at depth discontinuities. - Modeling the offsets removes the requirement of computing the expectation over the disparity/depth values to get a real valued disparity/depth value. However, intuitively it's not clear why the loss formulation is helping (even though the empirical results shows its effectiveness). The paper suggest that the Wasserstein based loss function helps to compute divergence measures between the predicted distribution and GT distribution when they may not have common supports. However, shouldn't the offset modeling be able to fix that? Is the modeling not powerful enough to fix the problem or is it an optimization issue that prevents it from fixing the problem? It would be useful to have an explicit discussion on why this loss formulation is required. - The paper has modeled the offset function as a function of its pixel coordinates u, v and disparity/depth d, i.e., we have a separate offset prediction for every u,v,d value. It would be interesting to see how well this approach would work if b is modeled as a constant function (i.e., same value for the entire image), function of pixel coordinates only, function of only d. It would also be interesting to qualitatively view the errors modes of different ways to model the offsets. - Are there any particular failure cases of the offset prediction, i.e., cases where the offset prediction has made the performance worse? - One of the main failure cases of stereo estimation are with thin structures, such as poles. Since they could have more than two modes in their prediction distribution, it might be better captured by this approach. It would be useful to report quantitative and/or qualitative results on such types of regions in the images. - Instead of simply reporting the stereo performance on edges (as done in Table 6), it would be interesting to report them on regions, i.e., pixel areas which are "close" to the depth discontinuity boundaries. Moreover, it would be important to report the numbers on the "interiors", to understand the relative impact of this approach. - Table 5 shows the impact of each contribution made by this paper on the depth estimation task. It would be useful to understand the impact of each of these on the downstream 3d object detection task as well. - Since using stereo estimation can be important for many downstream tasks, it would be beneficial to report the throughput of this approach. - It would be good to have the results of using the KL divergence approach in the main paper itself.

Correctness: - Yes, the claims and method discussed in the paper is correct. The design of experiments and the results align with the conclusions stated in the paper.

Clarity: - The paper has been written clearly. The reason behind the choice of this approach is very clear and the experiments results and their discussion are easy to be understood.

Relation to Prior Work: - The related work section is complete. This approach builds upon the current state of the art stereo estimation approaches. It explains clearly how this approach is different from the existing approaches and what missing aspect of the current approaches is being solved by the proposed solution.

Reproducibility: Yes

Additional Feedback: Final Review: I am not changing my initial review. This paper proposes a principled approach to train a stereo estimation problem, especially on regions of discontinuous depth.

[Author Response · NeurIPS 2020]

We thank the reviewers for their valuable feedback. We are encouraged they found our method well-motivated (R1, R2,
R3), rigorous (R1), novel (R2, R4), simply reproducible (R1) and effective (R3), compatible with other algorithms (R1,
R3), and well-validated by experiments (All). All the reviewers found our paper well-written and clear to follow. Given
the time and page limit, we respond to the major comments and will incorporate all feedback.

**@R1- GANet:** Given the limit time, we only managed to train and evaluate **CDN**-GANET Deep MM on Scene Flow:
0.68/7.7/2.97 (EPE/1PE/3PE) (cf. Table 1). We will include other results on GANET in the final version.
**@R1- Why multi-modal ground-truths (GTs)?:** There are three reasons. First, pixels are discrete: a single pixel may
capture different depths. Second, real datasets need to project signals from a depth sensor (e.g., LiDAR) to a depth
map. As pixels are discrete and the cameras and LiDAR might be placed differently, multiple LiDAR points of different
depths may be projected to the same pixel. Third, for stereo estimation, pixels along boundaries or occluded regions
cause ambiguity to the model; multi-modal GTs offer better supervision for training, especially in early epochs.
**@R1- Why not MM in Table 1?** We want to emphasize the gain by our algorithm design. We report the non-MM
results for a fair comparison with baselines which are mostly trained with uni-modal depths. We will specify this.
**@R1- MM in Table 3 & 4:** Conceptually our approach should improve, but we still evaluate using the (likely noisy)
uni-modal GTs. We conduct an analysis as in Table 6: w/ MM achieves 2.08/13.2/8.65, better than w/o MM.
**@R1- $\alpha$ & $k$:** Table B shows the errors with varying $\alpha$ & $k$ on Scene Flow using **CDN**-SDN-MM (cf. Table 4). A
smaller $\alpha$ leads to a larger error, which makes sense as it relies less on the GTs. After all, we attribute the small gain in
Table 4 to evaluation using uni-modal GTs. MM does improve convergence and depth on boundaries (see above).

**@R1, R2, R4- Offset network:** We will add details. It has 30K parameters, only $0.3\%$ w.r.t.
PSMNET. The novelty is in a single loss to jointly learn the offset and the main network.
**@R1- Bin sizes:** Our method outputs modes and needs (a) the bin containing the correct depth
to have the highest probability and (b) the offset to be accurate. A smaller bin size makes (a)
harder. A larger bin size makes (a) easier but makes (b) harder as the range of offsets gets larger.

Table A: MM ablation.

| $\alpha$ | $k$ | RMSE | ABSR |
|---|---|---|---|
| 0.8 | 3 | **1.80** | **0.028** |
| 0.8 | 5 | 1.82 | 0.029 |
| 0.8 | 7 | 1.88 | 0.035 |
| 0.5 | 3 | 1.81 | 0.029 |
| 0.2 | 3 | 2.20 | 0.062 |

**@R1- Kendall [12]:** 3D Convs smooth the estimation but cannot guarantee uni-modal distributions. [12] employs
pre-scaling to sharpen the probability (in their Fig 2), which might resolve the issue but makes the prediction concentrate
on discrete disparity values. We do not prevent predicting a multi-modal distribution, especially for pixels whose
disparities are inherently multi-modal. We output $\mathrm{argmin}$ (after an offset), which is what [12] hopes to achieve.
**@R1, R3- All Areas on KITTI:** There are two possible reasons. First, **CDN**-GANET overly focuses on foreground
pixels that contain more ambiguities and discontinuities. Second, we used the same hyper-parameters as the original
GANET and did not specifically tune it for **CDN**. We note that, # foreground:# background pixels is $\sim 0.15/0.85$; the
degradation on background is $\sim 0.16$ 3PE for both non occlusion and all, smaller than the gain on foreground.
**@R2- Learned offsets, explanations, insights:** Fig 3 shows how the offsets shift the distribution on a pixel and we
will add more qualitative results. The offset network learns to produce the sub-grid disparity at each grid disparity
values. The bin size balances the difficulty of predicting the bin location and the offset (please see @R1- Bin sizes) and
we found $s = 2$ to perform well. It is the only hyper-parameter to tune and only integral values are considered.
**@R2- KL divergence (KLD):** We apply the Wasserstein distance (WD) to overcome non-overlapped supports in
measuring divergences, which occur even if the target $p^\star$ is Dirac. Thus, using the WD is valid and more preferable
than manually adding a smoothing Gaussian/Laplacian to the KLD. While in Eq. (10) one can pair the offset with either
$\tilde{p}$ or $p^\star$, it makes more sense to view the offset as a way to improve the prediction $\tilde{p}$ rather than to adjust the target $p^\star$.
**@R2- Literature survey:** We will include more papers, especially those that discuss mean/mode and KLD.
**@R2- Ablation (cf. Table 5):** We use the mode for the WD-only model. Using a bin size $s = 2$ w/o offsets, the mode
is restricted to integers and EPE suffers. Using mean has 1.26/13.5/4.18, worse than mode since the WD does not align
the mean to the GT. Using mode for PSMNET has 1.57/39.7/4.40, worse than mean with a similar reason.
**@R3- CDN-SDN on KITTI:** We showed it in Table S3 (Suppl.). **CDN**-SDN is for depth estimation and we trained it
on KITTI detection following [43] (L244). See also Table B for the results on KITTI detection Val using other metrics.

**@R3- L272-276:** Our approach has advantage on hard pixels whose disparity is
ambiguous. We see (a) a gain on the foreground and (b) that foreground has a higher
error than All. We thus argue that most of these hard pixels are on the foreground.

Table B: CDN-SDN on KITTI.

| Method | RMSE | ABSR |
|---|---|---|
| SDN | 3.08 | 0.044 |
| CDN-SDN | 3.00 | 0.042 |
| CDN-SDN-MM | **2.99** | **0.040** |

**@R3- L299-300:** We visualized the depth results w/ and w/o MM at early epochs and
observed this. We will include both qualitative and quantitative results (like Table 6).
**@R4- Semantic segmentation:** Thanks for pointing out these papers that use semantic labels to guide the model to
resolve depth discontinuities (i.e., predict uni-modal distributions). Our method, in contrast, does not prevent predicting
multi-modal distributions along depth discontinuities, but changes the outputting rule (i.e., $\mathrm{argmin}$ with a predicted
offset). Our method can also capture depth discontinuities within an object or an object class.
**@R4- Modeling the offsets:** While the *learned* offsets may lead to common supports between the predicted and GT
distributions, we have to first come up with a loss to *learn* the offsets. Concretely, to learn $b$ in Eq. (10), we need a loss
that can measure the divergence between $\tilde{p}$ and $p^\star$. The WD offers a principled loss to learn the two networks jointly.
**@R4- Others:** Thanks for the great suggestions on analyses and we will try to include them in the final version.

[Meta-Review · NeurIPS 2020]

A method for stereo matching based on a Wasserstein distance loss is studied. All reviewers enjoyed the contributions of the paper. Quite a few suggestions to further improve the submission for the camera ready version were provided and the AC encourages the authors to take those into account for a more compelling read and presentation.